# Spectral Transmission of the Human Corneal Layers

**DOI:** 10.3390/jcm10194490

**Published:** 2021-09-29

**Authors:** Cristina Peris-Martínez, Mari Carmen García-Domene, Mariola Penadés, María Josefa Luque, Ester Fernández-López, José María Artigas

**Affiliations:** 1Anterior Segment and Cornea and External Eye Diseases Unit, FISABIO-Oftalmología Médica (FOM), Av. Pío Baroja, 12, E-46015 Valencia, Spain; m.carmen.garcia-domene@uv.es (M.C.G.-D.); mariola_penades_fons_@hotmail.com (M.P.); maria.j.luque@uv.es (M.J.L.); ester.fzlopez@gmail.com (E.F.-L.); Jose.Artigas@uv.es (J.M.A.); 2Surgery Department, Ophthalmology, School of Medicine, University of Valencia, Av. Blasco Ibáñez, 15, E-46010 Valencia, Spain; 3Department of Optics and Optometry and Vision Sciences, Physics School, University of Valencia, Dr. Moliner, 50, E-46100 Valencia, Spain; 4Thematic Cooperative Health Network for Research in Ophthalmology (Oftared), Carlos III Health Institute, C/Sinesio Delgado, 4, E-28029 Madrid, Spain; 5Pathology Group, PASAPTA, Veterinary School, Cardenal Herrera-CEU University, C/Tirant lo Blanc, 7, E-46115, Alfara del Patriarca, E-46115 Valencia, Spain

**Keywords:** transmittance, cornea, layer, ultraviolet, absorption coefficient

## Abstract

We have assessed the spectral transmittance of the different layers of the human cornea in the ultraviolet (UV), visible, and near-infrared (IR) spectral ranges. Seventy-four corneal sample donors were included in the study. Firstly, the corneal transmittance was measured using a spectrophotometer. Then, all samples were fixed for histopathological analysis, which allowed us to measure the thickness of each corneal layer. Finally, the absorption coefficients of the corneal layers were computed by a linear model reproducing total transmittance. The results show that corneal transmission was almost in unity at the visible and IR ranges but not at the UV range, in which the layer with higher transmission is Descemet’s membrane, whereas the stroma showed the lowest transmittance. Regarding the absorption coefficient, the most absorptive tissue was Bowman’s layer, followed by the endothelium. Variations on transmittance due to changes in the stroma, Bowman’s layer, or Descemet layer were simulated, and important transmission increases were found due to stroma and Bowman changes. To conclude, we have developed a method to measure the transmittance and thickness for each corneal layer. All corneal layers absorb UV light to a greater or lesser extent. The absorption coefficient is higher for Bowman’s layer, while the stroma is the layer with the lowest transmittance due to its thickness. Variations in stroma thickness or changes in the corneal tissue of Bowman’s layer or the endothelium layer due to some pathologies or surgeries could affect, to a greater or lesser degree, the total transmission of the cornea. Thus, obtaining accurate absorption coefficients for different layers would help us to predict and compensate these changes.

## 1. Introduction

The knowledge about the spectral transmission of the ocular media, the ratio between output and input, and output radiant flux under fixed geometrical conditions is important because it allows us to know the type and amount of radiation that reaches the retina [1,2,3,4,5]. This information is crucial because the cornea is the most anterior ocular medium that filters incident radiation [1,2,3] and, therefore, the first protection barrier against harmful radiation. Under normal conditions, the cornea blocks ultraviolet C (UVC) and practically all ultraviolet B (UVB) radiation, but it allows most of ultraviolet A (UVA) radiation to pass through [1], which implies that practically all the UVA radiation from the sun reaches the crystalline lens.

The cornea is formed by six different layers with different thickness and structure [6]. The epithelium is composed of five or six layers of uniform cells, reaching approximately 51 µm in thickness [7]. Epithelial cells are constantly renewed, so the main functions of this layer (barrier against fluids and microorganisms, physical protection against external traumas and refractive power) are permanently guaranteed.

Bowman’s layer is about 8–12 µm thick and is located right below the basement membrane of the corneal epithelium [8]. It is acellular and unable to regenerate once it is damaged [8,9,10]. It is considered to be an evolutionary adaptation because it is only present in some species, such as humans; other primates; and some birds, such as hens [11,12]. Although it has not yet been attributed a clearly defined function, it is postulated that it is determinant in functions as variable as protection against solar radiation, biomechanical stability, and correct corneal scarring, preserving its transparency after traumas [13].

The stroma is the central layer of the cornea, and its 500 µm represents 90% of the corneal thickness. The structure of the corneal stroma is made up of cells called keratocytes, extracellular matrix, and collagen fibers. In turn, the extracellular matrix is made up of proteoglycans and water. These components are found in different proportions, depending on whether it is the most anterior or posterior part of the corneal stroma. The anterior corneal stroma (anterior one third) has 30% more keratocytes than the posterior stroma; a greater crossover of collagen fibers; and a higher concentration of dermatan sulfate, a less hydrophilic proteoglycan than keratan sulfate. The latter, more hydrophilic, is found in greater proportion in the posterior two-thirds of the corneal stroma [14]. From a biomechanical point of view, this could impact the tensile forces of the corneal stroma [15]. These changes in stroma composition could affect the transmission of the stroma.

Dua’s layer is an acellular, strong layer in the pre-Descemet’s cornea with approximately 10 microns, composed of five to eight lamellae of predominantly collagen bundles [6]. Descemet’s membrane is the endothelial basement membrane, measures 3 µm in thickness at birth, and increases approximately 1 μm per decade [16]. Finally, the endothelium is a 5 µm-thick, single layer of hexagonal cells.

A first experimental ex vivo measurement of the spectral transmission of the human cornea was already carried out by Boettner and Wolter in 1961 [1] using a small number of corneas from donors of different ages to measure exclusively their total transmission, regardless of any possible difference among the different layers.

The measurements by Kolozsvári and co-workers have already revealed different spectral absorption characteristics of the three anterior corneal layers, concluding that Bowman’s is the layer with more absorption, and the stroma is the layer with less UV transmission [17]. However, it would be important to study the spectral transmission of the different layers of the cornea throughout the spectrum, i.e., ultraviolet, visible, and infrared. This would give us information about the transparency of the cornea, as well as about the type of radiation that reaches the crystalline lens according to the layers and thickness of each specific cornea.

This information is crucial for patients who undergo surgical procedures that totally or partially remove some corneal layers. In addition, knowing the transmittance of the normal corneal layers can help us to know the influence of ultraviolet absorption in patients with certain corneal pathologies or after some surgical procedures. For instance, corneal diseases such as keratoconus, which may result in a progressive thinning of the corneal and cells composition [18], might affect corneal spectral transmission, and it would be important to protect the cornea, the lens, and the retina from the UV radiation that would normally be blocked by a healthy cornea.

In this paper, we experimentally determined the spectral transmission of the different layers of the human cornea, as well as its total transmission between 280 and 1100 nm, i.e., for UVA, UVB, visible, and infrared A (IRA).

## 2. Materials and Methods

### 2.1. The Samples

The donated samples were collected within 6 hours of the donor’s death. They consisted of a corneo-scleral ring measuring approximately 20 mm in diameter, stored in EUSOL-C dextran corneal storage media (Alchimia, Ponte S. Nicolò, Padova, Italy), at 4 ºC. Evaluation of the corneas was carried out following the criteria of the European Eye Banking Association (EEBA) [19,20]. For this purpose, a slit lamp and a specular microscope Konan CellChek EB-10 (Konan Medical USA, Inc., Laguna Canyon Rd, Irvine, CA, USA) were used. Of each sample, only the layers meeting the following inclusion criteria were included:

### 2.2. Inclusion Criteria

Corneas from donors that were discarded for transplantation (due to infections by hepatitis B or C viruses, HIV, or Toxoplasma) or the excess corneal layers from partial transplants which, after 10 days of storage (established by the EUSOL-C^®^ protocol), have not been claimed by another hospital of the Valencian Community to be used for another transplant.Layers of the cornea that meet the quality criteria of the EEBA.

### 2.3. Exclusion Criteria

For complete corneas:○Corneas stored in EUSOL_C for more than 11 days.○Abnormal corneal shape (due to keratoconus, micro- or megalocornea), signs of prior surgery in the anterior segment, signs of any refractive surgery, signs of inflammation.For individual layers:○Epithelium: epithelial irregularities affecting the number or the morphology of the epithelial cells.○Bowman’s layer: structural abnormalities.○Stroma: diffuse thickening; edema; presence of focal opacities, scars, folds, grooves, or other pathological changes.○Descemet’s membrane: denuded membrane, presence of folds or grooves.○Endothelium: precipitates, corneal guttae, scars, folds, grooves, severe polymegethism, severe pleomorphism, severely damaged cells (necrotic/apoptotic), dead cells (resulting, for example, from trauma or post-mortem cell decay, etc.), or cell counts <1500 cells/mm^2^.○Abnormalities detected by histological analysis after sample processing.

### 2.4. Transmittance Measurement

For this study, a central button of each cornea was cut out using a 7 mm diameter trephine. This area is selected in order to measure the central 5.5 mm, since it is the average optical zone used both in day vision (3 mm pupils) and in low-lighting conditions (pupils of around 5 mm). A slightly larger button avoids the possible artifacts that could occur if the edge of the sample was included in the measurement.

If the received cornea was complete, its transmission was measured in order to correct the disrupting influence of the dye in the corneal transmission. After the measurement, Descemet’s layer and the endothelium layer were separated from the rest. To that aim, they were dyed and visualized in the microscope. A drop of VisionBlue (Trypan Blue Ophthalmic Solution DORC Dutch Ophthalmic Research Center, Zuidland, The Netherlands), a biocompatible and non-toxic dye for corneal cells, was used. We measured the transmittance of both Descemet’s layer and/or the endothelium. The endothelium cannot be measured by itself since Descemet’s layer is its basement membrane, and when they are separated, it does not remain intact. On the contrary, Descemet’s membrane can be found lacking the endothelial cells.

Regarding the epithelium, an efficient separation technique to obtain a layer suitable for measurement has not yet been developed. Therefore, in some of the corneas, the three anterior layers were measured together, whereas in others, the epithelium was removed by scraping, and Bowman’s layer and the stroma remained together. It was neither easy nor accurate to separate Bowman’s layer from the stroma manually, so a microkeratome EVO3D (MORIA Ophthalmic instruments, Antony, France) was used, leaving as little stroma as possible attached to Bowman’s layer.

We do not have a technique to separate the stroma and Dua’s layer. Therefore, in the Results and Discussion sections, the term “stroma” refers to “stroma and Dua’s layer” together.

During the entire preparation process of the corneal layers, the sample was wetted with balanced saline solution (BSS plus, Alcon Laboratories Inc. Fort Worth, TX, USA) in order to preserve the hydration of the cornea. Before measuring the transmittance, the sample was washed in BSS for 10 minutes, changing the BSS after 5 minutes to eliminate the storage media.

The different layers obtained were placed between two sapphire sheets 7.5 mm in diameter and 0.4 mm thick, forming a sandwich, which, in turn, was placed on a caliper specially designed for this purpose, measuring 5.5 mm in diameter (see Figure 1). This caliper fits perfectly into the inlet opening of the integrating sphere of the spectrophotometer used (Perkin-Elmer LAMBDA 35 UV/VIS. PerkinElmer, Waltham, MA, USA).

This device measures the spectrum from 280 to 1000 nm, which means that spectral transmissions in UVB, visible, and IRA are accurately determined. The measurement precision is up to 5 nm for wavelength and up to 2% for transmittance. The sapphire sheets in air were taken as a blank reference to measure transmittance. Measurements were not corrected for reflection in the first surface of the corneal tissues, and, therefore, we work with the measured and not the internal transmittance. We have taken the transmission measurements with an integrating sphere—a hollow sphere with a high reflectance coating that collects all the light transmitted by the sample—and the measures refer to the total transmission and not only to the direct component. The mean time to perform the measurement was 2 minutes. After the measurement of the transmittance, each sample was prepared for further histological analysis.

### 2.5. Fixing, Processing, and Histological Analysis of Samples

Tissues were fixed in 4% neutral buffered formalin (Química Clínica Aplicada S.A., Amposta, España) and dehydrated through graded alcohols before being embedded in paraffin wax. Several 4 µm thick sections were cut from each sample using a manual microtome (RM2235, Leica Microsistemas S.L.U., Barcelona, Spain) and were routinely stained with hematoxylin and eosin (HE). Then, all sections were evaluated using the light microscope with an attached camera that was able to take scaled photographs of the layers of each corneal sample. At this stage, a pathologist looked for possible tissue abnormalities caused by the measurement process that would lead to the sample being discarded. ImageJ was used to determine the exact thickness of each layer. To do this, five measurements of the thickness of each corneal layer were taken, randomly distributed along the histological section. According to Kolosvàry et at. [17], formalin could dehydrate the tissues, and this has been taken into account; we have introduced a +5% correction in each layer thickness. Finally, the average measure for each layer was calculated.

### 2.6. Calculation of the Absorption Coefficient

The spectral transmittance of a transparent sample, *T_λ_*, defined the ratio between the transmitted and incident light for a given measurement geometry for wavelength *λ*, which is related to sample thickness, *d*, by the Beer–Lambert law (Equation (1)) [21]:(1)$$T_{\lambda }=t_{\lambda }{}^{d}$$
where *t_λ_* is the transmittance for unit thickness. Absorbance, *A_λ_*_,_ is defined as the decimal logarithm of the inverse of the transmittance and is a measurement of how the sample attenuates light (because of absorption or scatter, for instance) [21]. From (1) it follows that
(2)$$A_{\lambda }=-log_{10}T_{\lambda }=-d\times log_{10}t_{\lambda }=d\times \alpha_{\lambda }$$
*α_λ_*, the absorbance per unit thickness, is usually called the absorption coefficient.

The transmittance of a corneal sample (*s*) can be computed as the product of the transmittance of its layers, and, therefore, the following formula would hold for the corneal spectral absorbance:(3)$$A_{s,\lambda }=-log_{10}T_{s,\lambda }=\overset{5}{\underset{j=1}{\sum }}d_{s,j}\times \alpha_{j,\lambda }$$
where labels *s* and *j* identify, respectively, the corneal sample and the layer number (*j* = 1 epithelium, *j* = 2 Bowman’s layer, *j* = 3 stroma, *j* = 4 Descemet, and *j* = 5 endothelium), and where missing layers have zero thickness.

Once the transmittances for each sample (*T_s_*_,*λ*_) and the thicknesses (*d_s,j_*) of each layer were measured, according to Equation (3), the absorption coefficients for the different layers can be obtained by linear regression. Least-squares fitting by Matlab (version R2019a. The MathWorks Inc., Natick, MA, USA) software has been used to this end, using all samples with one or more layers to train the mathematical model. The average correlation coefficient (aCC) [22] between the experimental and theoretical log-transmittance values obtained for each wavelength was used as a measurement of the model’s performance. From the absorption coefficients and Equations (2) and (3), the predicted transmittances of individual layers and complete corneas were computed to test the model’s predictions.

## 3. Results

Seventy-four samples from 45 cornea donors were included in the study. The mean age was 64 ± 13 years, and 42% of the participants were women while 58% were men. The storage time until sample processing was 6 ± 3 days.

The analyzed samples were distributed as follows: 6 including epithelium + Bowman’s + stroma + Descemet’s, 8 with epithelium + Bowman’s + stroma, 26 with Bowman’s + stroma (with different stroma thickness: from 47 μm to 594 μm, total stroma), 1 with only Bowman’s layer, 3 with only Stroma, 4 of Bowman’s + stroma + Descemet’s, 11 with Descemet’s, and 15 with Descemet’s + endothelium. Table 1 shows the data of the different layers, number of samples of each layer, and their corresponding mean thickness.

These corneal samples had been discarded for transplant due to the following different causes: 30% due to diffuse degeneration invalidating the endothelial cell counting, 24% due to low endothelial cell counting (between 1500 and 2000 cells/mm^2^), 20% excess from partial Descemet’s + endothelial transplant, 11% due to presence of guttas, 9% exclusion due to donors infected by hepatitis B virus, and 6% due to presence of peripheral leucoma. Only the individual corneal layers of each sample verifying the inclusion–exclusion criteria were used; the rest were discarded.

The absorption coefficients of each corneal layer were obtained by linear regression using Matlab software, as described in the Methods section. The fit of this model was good, with an average correlation coefficient (aCC) of 0.76. The results are shown in Figure 2. As can be observed, absorption occurs mainly at the UV range. When comparing the different layers, it is remarkable that the most-absorbing layer was Bowman’s layer. From 280 to 400 nm, the layers differed in behavior. The most-absorbing one was Bowman’s, whereas the least-absorbing layer was the stroma. From 400 nm onwards, the absorption was mostly uniform. The UV absorption (280–400 nm) of different layers ranked as follows: Bowman’s, endothelium, stroma, epithelium, and Descemet’s.

Finally, three situations likely to cause changes in the corneal transmission have been calculated and represented: 1) the decrease in the stromal thickness, 2) the removal of Bowman’s layer, and 3) changes in the thickness of Descemet’s layer due to aging. Regarding the human stroma, different thicknesses were evaluated (400 and 500 μm) assuming that the absorption coefficient is the same for the whole stroma. This assumption, taking into account the slightly different composition of the stroma according to the area analyzed (anterior or posterior), must be interpreted with caution. For every 100 μm of thickness’ stroma loss, there is an increase of 7% in the transmission at UV range. As we can see in Figure 3a, the reduction in the stroma layer thickness raises the total corneal transmission, increasing it more than 5% (from 320 to 400 nm) for the removal of 100 μm of stroma. Figure 3b evidences that if Bowman’s layer is eliminated, the transmission increases more than 10%, from 315 nm to 400 nm. The changes in total cornea transmission due to the thickness increment in Descemet’s layer throughout life are less than 1% at UV range.

## 4. Discussion

This study develops a new methodology to determine the absorption coefficients of different donor human corneal layers based on transmittance and thickness measurements. For the first time, we have obtained transmittance data from Descemet’s membrane and endothelium samples. All corneal layers offer a practically total transmission at visible and near-IR range, but at UV range (from 280 to 400 nm), several layers present differences in absorption (see Figure 2), with the highest absorption coefficient corresponding to Bowman’s layer and the layer with less transmission to the stroma, due to its greater thickness. However, the stroma did not show the highest coefficient of absorption. In this section, we will compare these results with the previous literature, whenever possible. We have not found transmittance or absorption results for endothelium or Descemet’s membrane in humans.

Results are conditioned by the age range of the donors (64 ± 13 years) and the limitation in the number and type of available corneal layers, since the priority is to use them for transplantation. Due to the difficulty encountered in separating completely isolated layers that could be measured (one Bowman’s and three stroma), the results should be interpreted carefully. Given that, in this study, we have not been able to separate Dua’s layer from the stroma, its influence on transmittance is embedded in the stromal data.

Care has been taken to ensure that the tissues included in the study were healthy and that dehydration was minimized. Endothelial cell density, however, was below 2000 cells/mm^2^ in some samples (values ranged from 1830 cells/m^2^ to 2793 cells/m^2^, with an average value of 2239 cells/m^2^). Although there is a lack of agreement in the literature on the normal ranges of endothelium cell density, which depend on age, ethnic group, and measurement device [23], it cannot be discarded that samples with low cell densities might bias the results.

### Comparison of the Obtained Results with Other Similar Studies

First of all, the measurement of the transparency of the cornea was obtained, and the spectral transmission of the cornea was compared with that obtained by Boettner and Wolter in 1962 [1]. The range of wavelengths used in our experiment is the broadest allowed by the spectrophotometer used, ranging from 280 nm (UVB) to 1000 nm (IRA), which is comparable with the results found in a related bibliography [1], although not reaching the far infrared. In that regard, Figure 4 shows the total spectral transmission of the human cornea obtained by Boettner and Wolter [1] and the total spectral transmission of the cornea obtained in this study. For this purpose, a theoretical total corneal transmittance was calculated, taking into account the parameters of an average cornea (50 μm for epithelium, 500 μm for stroma, 10 μm for Bowman’s layer, 9 μm for Descemet’s layer, and 5 μm for endothelium) [7,8,14,16].

The shapes of both curves are very similar. The small differences found in transmission could be due to the different age of the donors used in both studies. Younger donors (from 4 weeks to 75 years of age) included in the Boettner and Wolter study seem to have more transparent corneas than the adult donors in our study (64 ± 13 years) [1].

The data obtained for the different layers that make up the cornea can be compared with those obtained by Kolozsvári et al. [17]. This comparison can only be made for the UV range, as these authors determine the absorption coefficients of the different layers of the human cornea only for the 240–400 nm range. This range is important because UV radiation is potentially the most dangerous for the inner structures of the eye. Although Kolozsvári et al. [17] determine experimentally the absorption coefficients of the different layers, and we estimate them from spectral transmittance, these data are comparable, qualitatively speaking, since the most-absorbing layers logically coincide with those with lower transmittance, although thickness must be taken into account to determine the total transmittance. Since the numerical values from Kolozvári et al.’s study were estimated from the plots, it is difficult to draw reliable conclusions. The spectral transmittance of the stroma and the whole cornea between 240 and 280 nm is nearly zero, and the epithelium and Bowman’s layer are around 20%. Moreover, above 300 nm, the stroma and cornea curves on one side and the epithelium and Bowman’s layer curves on the other side are virtually indistinguishable.

However, it is possible to deduce from the results of Kolozsvári et al. [17] the value of the absorption coefficients (α) of the different layers of the human cornea, which is possible because the ordinate axis is in linear scale. If we compare these results with ours (Figure 5), it can be seen that, although the results are not identical, the shape of the curves and the coefficient values are very similar for the three layers. Thus, in the spectral regions where they can be compared, our results are consistent with those of Boettner and Wolter [1] and Kolozsvári et al. [17]

## 5. Conclusions

The most absorptive corneal tissue for the UV light is Bowman’s layer, followed by the endothelium, while the stroma is the layer with less transmission due to its thickness. In the spectral transmission of the corneal layers, it is as important to take into account the absorption coefficient of the tissue as the thickness of the analyzed layer. Variations in the thickness or in the absorption of the corneal layers could affect, to a greater or lesser degree, the total transmission of the cornea. In future studies, it would be important to determine the effect of age on corneal transmittance to compare the transmission between anterior and posterior stroma and to measure Dua’s layer transmittance.

As a clinical application, we want to analyze the influence of surgical procedures, such as corneal transplant and refractive surgery, and of certain pathologies, such as keratoconus or Fuchs’ endothelial dystrophy [24]. Depending on these results, it could be important to prescribe a UV filter in order to protect the cornea, the lens, or, finally, the retina. Moreover, our results might help in cell phototherapy [25].

## Figures and Tables

**Figure 1 jcm-10-04490-f001:**
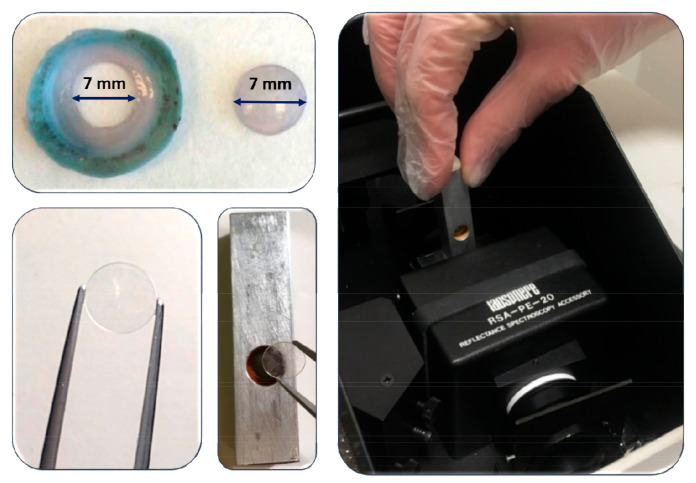
Process of transmittance measurement: corneal sample preparation using a 7 mm diameter trephine, placement between sapphire sheets and then inside the sample caliper and inside the spectrophotometer.

**Figure 2 jcm-10-04490-f002:**
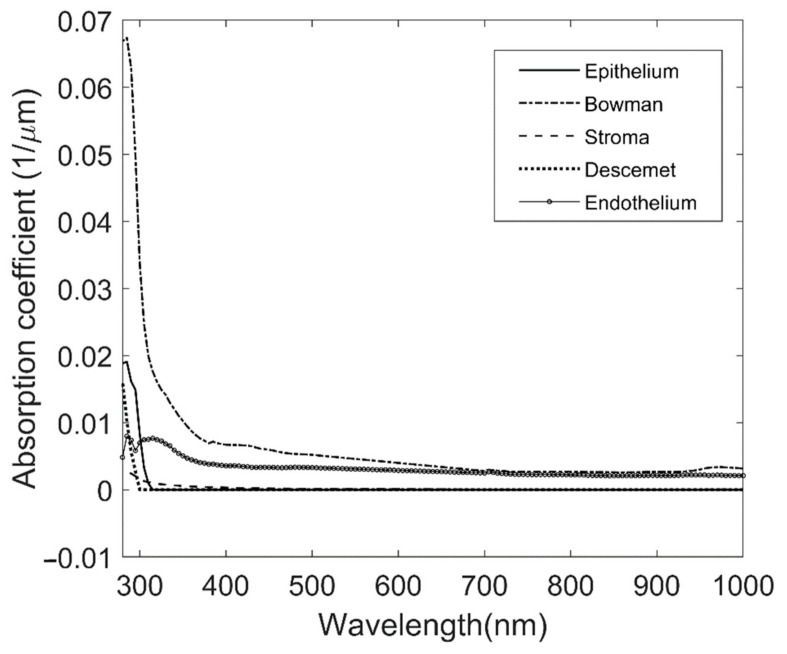
Absorption coefficient for the five corneal layers for all the measured ranges.

**Figure 3 jcm-10-04490-f003:**
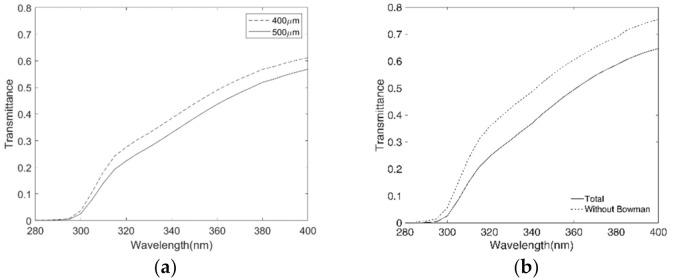
Changes in transmittance in the ultraviolet A+B region for: (**a**) total cornea with two stroma thicknesses (400 μm and 500 μm) and (**b**) total cornea with and without Bowman’s layer. Data of average thickness of each layer in a normal cornea have been used for this simulation: 50 μm for the epithelium, 10 μm for Bowman’s layer, 500 μm for stroma, 9 μm for Descemet’s layer, 5 μm for the endothelium [7,8,14,16].

**Figure 4 jcm-10-04490-f004:**
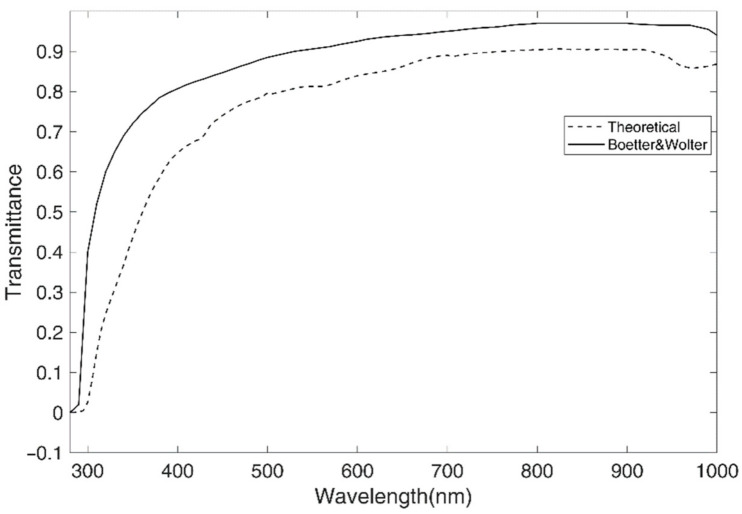
Comparison of the Boettner and Wolter [1] experimental result of total transmittance cornea with our calculated results from the absorption coefficients and mean thickness of corneal layers.

**Figure 5 jcm-10-04490-f005:**
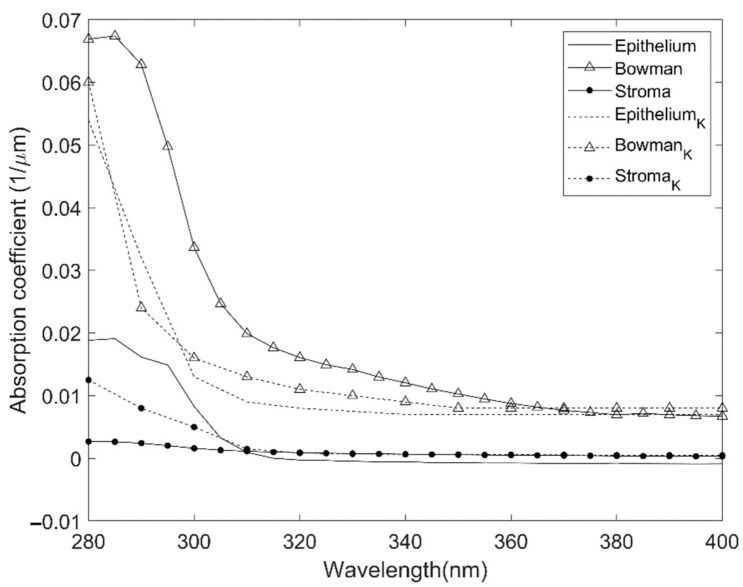
Comparison of our absorption coefficient results for epithelium, Bowman’s, and stroma with those obtained from Kolosvári et al. [17] epithelium_k_, Bowman_k_ and stroma_k_.

**Table 1 jcm-10-04490-t001:** Data of corneal layers. Number of samples of each layer (N), mean thickness (T), and standard deviation (SD).

	Epithelium	Bowman	Stroma	Descemet	Endothelium
N	14	45	47	36	15
T ± SD (μm)	20 ± 12	8 ± 2	414 ± 111	6 ± 2	7 ± 2

## Data Availability

Data available on request due to restrictions of privacy.

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
