# Peer review of "Spectral Transmission of the Human Corneal Layers"

_jcm, 2021, doi:10.3390/jcm10194490_

Round 1

Reviewer 1 Report

In this manuscript, the authors present a method and results for determining the transmittance and absorption coefficients of the different corneal layers in the ultraviolet, visible, and infrared spectral ranges. For this purpose, a total of 74 corneal samples were measured by spectrophotometry. Because the samples consisted of whole corneas but also a few layers and the layer thicknesses of all samples were measured, the transmittance and absorption coefficients of each layer could be determined by linear regression from the measurement.

This work provides an important contribution, as previous work only determined the transmission of the entire cornea or only in the UV range for some layers but not all. The results of this manuscript are compared with those of other works and it is additionally shown which influences a structural change of the cornea has on the transmittance in the UV range. As the authors state, their results could make an important contribution regarding UV exposure of the crystalline lens or retina in various diseases or after certain surgical procedures.

However, the manuscript needs some major and minor revisions. The methods, while well and soundly presented, should be more detailed for readers unfamiliar with spectrophotometry.  In all figures concerning the absorption coefficient, the unit is missing. This must be revised. Furthermore, a “Table 1” is mentioned in the text, but it is nowhere to be found in the manuscript.

Further detailed comments follow with line references:

Line 54-56 – “The composition of the corneal stroma is not homogenous, since the characteristics of its anterior and posterior regions (one third and two thirds, respectively, of the whole layer) are very different [14].“
Please specify the differences and discuss their influence on the transmittance.

Line 133-138 – “If the received cornea was complete, the Descemet and the endothelium layers were separated from the rest. To that aim, they were dyed and visualized in the microscope. […] Hence, the transmission of each cornea was previously measured in order to correct the disrupting influence of the dye in the corneal transmission.”
This part was confusing on the first read. The authors should rearrange the statements and first mention that the transmission of each cornea was measured.

Line 169-175 – In this part are some important details missing: Is the measurement reflection corrected? Did the authors consider the influence of the sapphire transmittance or did they use the sapphire sheets also for the blank reference? Please explain briefly for readers unfamiliar with spectrophotometry what an integrating sphere is.

Line 178 – Does dehydration have an influence on the layer thickness and thus on the results?

Line 191 & 198 – The authors should reference applicable literature for their formulas.

Line 215 – “Table 1 shows the data of the different layers, […]”
Where is Table 1?

Line 217-221 – The authors should discuss whether the reason for excluding the corneas for transplantation might have an impact on their measurement, for example, low cell density in the endothelium.

Line 221-222 – “The absorption coefficients of each corneal layer were obtained by linear regression using Matlab® software.”
This sentence belongs in the Methods.

Line 223-224 – How did the authors calculate this R² coefficient? As I understand, the linear regression was applied over the entire spectrum or in other words, for each wavelength. Thus, there should also be an R² for each wavelength. Could you explain how you calculate only one value for the entire spectrum?

Line 232 – The unity of the absorption coefficient is missing in the figure.

Line 233 – “Figure 2. Absorbance coefficient for the five corneal layers for all the measured range.”
This should be “Absorption coefficient…”.

Line 241-242 – “Every 100 μm of thickness’ loss, there is an increase of 7% in the transmission at UV range.”
Please clarify that the 7% increase is referring to the stroma only.

Line 256-258 – Aren’t there other techniques to measure corneal layer thickness with sufficient accuracy?

Line 295-297 – “We were unable to extract accurate transmission values from Kolozvári study because the authors plotted their results on a logarithmic scale, so it’s difficult to draw reliable conclusions.”
I don’t see why this is a problem. The authors could just use Figure 5 from the Kolozvári study because it shows the spectral absorbance with a linear ordinate axis. Since the transmittance is the logarithmic and inverse of the absorbance it can directly be deduced from it. The authors should also consider plotting their results on a logarithmic scale.

Line 325-326 – “Author Contributions: For research articles with several authors, a short paragraph specifying their 325 individual contributions must be provided. The following statements should be used Conceptualization, all authors; […]”
Is this a leftover from a template?

Reviewer 2 Report

General comments

I am very glad that the authors wrote this article. The results of this study represent a significant contribution to understanding and also being able to act on different situations or pathologies that affect the cornea. Therefore, I encourage the authors to follow this line of research and thus be able to provide tools with clinical involvement.

However, some issues still need to be addressed before the editor can consider publishing the manuscript:

Introducction

I miss the definition of the coefficients of absorption and transmittance, as well as the relationship between the two concepts.

Perhaps, in order not to extend too much in this section, the authors could consider reducing somewhat what they mention about Boettner & Wolter and Kolozsvári and co-workers.

Materials and methods

The authors suggest that the maximum time from the extraction of the donor cornea and the transmittance study was not greater than 11 days. This is so? I think at least the authors could mention the mean time in which the measurements were made on the specimens. This data can be included in section 2.4 (2.4 Transmittance measurement).

Results

Table 1 (line 215) is mentioned but does not appear in the text.

Discussion

In figure 5, in the upper right box, there is a typographic error (where it says EnpitheliumK should put EpitheliumK).

References

In reference 13, Journal of Morphology should be replaced by J Morphol

Reviewer 3 Report

The authors of the manuscript titled  Spectral Transmission of the Human Corneal Layers experimentally determined the spectral transmission of the different layers of the human cornea, as well as its total transmission between 280 and 1100 nm, i.e. for UVA, UVB, visible and infra-red A (IRA).

I appreciate the novelty of such approach, since many ophthalmic surgical procedures affect corneal layers separately without described effect on the light transmission.

The results, which I find especially useful is that the reduction of the stroma layer thickness rises the total corneal transmission, increasing it more than 5% (from 320 to 400 nm) for the removal of 100 μm of stroma. The second particularly important result is the evidences that if the Bowman’s layer is eliminated, the transmission increases more than 10%, from 315 nm to 400 nm.

Few issues, which needs to be addresses:

  1. I got confused with the exclusion criteria from the study group including Endothelial Cell counts <2.000 cells/mm2. Yet, in the results section such corneas “These corneal samples had been discarded for transplant due to the following different causes: 30% due to diffuse degeneration invalidating the endothelial cell counting, 24% due to low endothelial cell counting (between 1500-2000 cells/mm2)” were included. Please clarify the inclusion and exclusion criteria for the study? Did only good quality corneas “normal subjects” were studied ?
  2. Please list limitations of the study in the discussion part.
  3. Please add important future directions of the stud, such as: age groups, corneal after refractive surgery procedures - Bowman layer sparing procedures versus PTK

Round 2

Reviewer 1 Report

Thank you for the detailed response to my questions and the revision of the manuscript.

Author Response

Thanks to you for the suggested corrections, they have helped us to improve the manuscript considerably.